# Electrochemical Detection and Point-of-Care Testing for Circulating Tumor Cells: Current Techniques and Future Potentials

**DOI:** 10.3390/s20216073

**Published:** 2020-10-26

**Authors:** Chunyang Lu, Jintao Han, Xiaoyi Sun, Gen Yang

**Affiliations:** State Key Laboratory of Nuclear Physics and Technology, School of Physics, Peking University, Beijing 100871, China; 1701110253@pku.edu.cn (C.L.); 1801110241@pku.edu.cn (J.H.); 1901110259@pku.edu.cn (X.S.)

**Keywords:** circulating tumor cells, electrochemical detection, point-of-care testing

## Abstract

Circulating tumor cells (CTCs) are tumor cells that escaped from the primary tumor or the metastasis into the blood and they play a major role in the initiation of metastasis and tumor recurrence. Thus, it is widely accepted that CTC is the main target of liquid biopsy. In the past few decades, the separation of CTC based on the electrochemical method has attracted widespread attention due to its convenience, rapidness, low cost, high sensitivity, and no need for complex instruments and equipment. At present, CTC detection is not widely used in the clinic due to various reasons. Point-of-care CTC detection provides us with a possibility, which is sensitive, fast, cheap, and easy to operate. More importantly, the testing instrument is small and portable, and the testing does not require specialized laboratories and specialized clinical examiners. In this review, we summarized the latest developments in the electrochemical-based CTC detection and point-of-care CTC detection, and discussed the challenges and possible trends.

## 1. Introduction

Cancer is one of the leading causes of death, and around 90% cancer death due to metastasis [1,2]. Therefore, achieving an earlier cancer diagnosis is of fundamental importance. For conventional needle biopsy techniques, the invasiveness limits its use. Meanwhile, liquid biopsy techniques analyze tumor cells or tumor cell debris from blood or other body fluids, including circulating tumor cells (CTCs), circulating tumor DNA (ctDNA), extracellular vesicles (EVs), and exosomes, etc. [3]. Compared with needle biopsy, its non-invasiveness allows us to collect patient blood samples continuously, and to realize real-time monitoring of patient disease progression and personalized medicine [4,5]. Moreover, since CTC, ctDNA, EVs, etc. can be released from both primary and metastatic tumors, liquid biopsy provides us more comprehensive information [6,7]. CTC is the main target of liquid biopsy, for CTC is the most important part during the metastasis process [8]. It has been reported that CTC could be detected before cancer forms metastasis [9,10]. Detection of CTC in the blood could be used to achieve an earlier diagnosis and a better control of cancer, and avoid the bad consequences caused by cancer metastasis. Besides, CTC could be used to assess the patient prognosis and evaluate the treatment outcome in real-time [5,11]. The isolation, culture, and sequencing of CTC could also help us to determine patients’ drug resistance and find potential therapeutic targets [12,13,14,15].

Separating CTC from complex blood components is extremely challenging. The amount of CTC in the blood is extremely rare, in average about 1–100 mL^−1^ [16], while the number of white blood cells (WBC) and red blood cells (RBC) is about 0.4–1 × 10^7^ mL^−1^ and 3.5–5 × 10^9^ mL^−1^, respectively. The separation mainly relies on the difference in biological properties or physical properties between CTC and blood cells [17]. Biological properties-based CTC separation, mainly using the unique antigen expression on the surface of CTC, such as the most commonly used anti-epithelial cell adhesion molecule (EpCAM) and Cytokeratin (CK), etc. [18]. Coupling these antibodies to the surface of magnetic beads or the chip can achieve the specific capture and separation of CTC. Physical properties-based CTC separation mainly uses the difference in cell density, size, and deformability between CTC and blood cells to achieve CTC separation [19,20,21]. Although there are many CTC detection methods, their complicated operation process, high cost, and low sensitivity are still problems. In recent years, a lot of electrochemical methods have also been used to detect CTC, using aptamers and nanomaterials to modify the electrode, by recording the current change/electrical impedance spectrum change, and establishing a linear relationship between the change and the number of CTC to realize the quantification of CTC [22,23]. This ensures high sensitivity and selectivity, and has outstanding advantages, such as rapid response, easy operation, affordability, and nondestructive analysis [22].

After completing the separation of CTC, it is very important to quantify the number of CTC. In general, traditional biological properties-based methods and physical properties-based methods use fluorescently labeled antibodies to identify and count the captured CTCs. CTCs were recognized as Hoechst+ (nuclear dye), EpCAM+/CK+, and CD45- (WBC specific marker) cells, while WBCs were recognized as Hoechst+, EpCAM-/CK-, and CD45+ cells [24,25,26]. This fluorescence imaging-based method usually requires a specialized fluorescence microscope, which is expensive, and the output of the results requires professional technicians and also takes a long time, thus limiting its clinical use. The electrochemical-based method only requires some simple instruments, such as current meters, and the whole detection process is relatively simple and fast [27,28,29,30]. However, the preparation process of the device is complicated and the detection time is long. There is an urgent need for a simpler, more efficient, and faster method. Point-of-care testing (POCT) realizes target quantification through pressure, distance, color, etc. As a sensitive, fast, cheap, easy-to-operate method, it allows patients to realize sample input and result output, and it doesn’t need for complex equipment [31,32]. The test can be performed at the bedside of the patient and the results are available immediately. It is of great significance for popularizing the clinical application of CTC detection.

This review first briefly introduced biological properties-based and physical properties-based CTC detection methods, and then summarized the latest progress in the electrochemical detection of CTC. Further, we discussed advantages and disadvantages in the electrochemical detection of CTC and proposed possible development directions. Then, we introduced POCT and showed some examples. Finally, we discussed some possible combinations of POCT and CTC detection.

## 2. CTC Separation Methods Based on Biological Properties and Physical Properties

Various CTC enrichment technologies have been developed, which can be roughly divided into two categories, namely biological properties-based methods and physical properties-based methods. These methods usually combined with the microfluidic chip, the reason is that the microfluidic chip usually integrates various functions, including preprocessing, mixing, separation, detection, and so on. By reasonably designing channels, columns and chambers, cells could be separated well. In addition, it achieved high-throughput detection, hundreds of samples can be analyzed within a few minutes. The most important thing is that the microfluidic chip requires less reagent consumption, the cost is low, and it’s easy to operate [33]. We summarized the biological properties-based methods and physical properties-based methods in Table 1, and their application flow rate, capture efficiency, and purity were listed.

### 2.1. Methods Based on Biological Properties

Tumor metastasis includes four processes. (1) Invasion: tumor cells undergo epithelial-mesenchymal transition (EMT), which promotes the downregulation of adhesion between cells, and cells fall off from the primary tumor, obtain high mobility and invasiveness at the same time. (2) Intravasation: tumor cells pass through basement membrane (BM) and enter the blood vessels or lymph vessels, also called circulating tumor cells (CTC, including single CTC and CTC cluster). Only a very small number of CTCs can survive and form metastases. The reason for this is that CTCs are exposed to high shear stress in the blood and are attacked by immune cells [34]. They undergo apoptosis and are quickly eliminated. In addition, because the tumor cells leave the extracellular matrix (ECM), anoikis will occur [35]. Overall, CTCs have a short half-life in blood (1.0–2.4 h in patients with breast cancer) [36]. (3) Extravasation: when the CTC reaches the distal capillaries (diameter: 6–9 μm), it may get stuck and get out of the blood or lymphatic vessels. This process relies on neutrophils to release matrix metalloproteinases (MMPs) to weaken the connection between endothelial cells. (4) Metastases: tumor cells may remain in dormant state for a period of time, when the external environment is suitable for proliferation, they may undergo mesenchymal–epithelial transition (MET), which allows cells to restore high adhesion properties, promoting them to establish connections, and finally form metastasis (Figure 1A) [10,37]. From the above description, we know that CTC plays an important part in the metastasis process. For different cancer types, CTC markers are slightly different. For example, in breast cancer, the commonly used markers are CK19, EpCAM and hMAM [38]; in lung cancer, the markers are CK7, CK19, and TTF-1 [39,40]; in prostate cancer, the markers are PSA and PSMA [41]; in colorectal cancer, the markers are CK and CD133 [42,43]; in pancreatic cancer, the markers are CK19, EpCAM, and MUC1 [43]; in ovarian cancer, the markers are EpCAM and MUC1 [44]. We showed the typical fluorescence image and SEM image of CTC (Figure 1B,C).

Biological properties-based CTC separation, usually using the unique CTC molecular markers, the most frequently used are EpCAM and CK, they are not expressed on blood cells. CellSearch^TM^ is the only platform approved by the FDA for commercial detection of CTCs. CTCs are combined with EpCAM-coated magnetic beads and then separated under the magnetic field. After fluorescent labeling, CTC is defined as Hoechst+, EpCAM+/CK+ and CD45- (Figure 2) [45,46]. Such magnetically activated cell sorting is called positive sorting [47,48]. Moreover, antibody-coated magnetic beads can also be bounded to the background cells, for example, using CD45 labeled magnetic beads to identify and remove WBCs, this kind of cell sorting is called negative sorting [49]. After that, researchers introduced this principle to microfluidic chips, and achieved the capture of CTC by coupling antibodies to the surface of the chip [24,25]. Some researchers have modified the surface of the chip with nanomaterials or surfactant/lipid bilayers, etc. [50] to promote the interaction between CTC and antibody and reduce non-specific cell adsorption [51], and effectively increase the CTC capture efficiency as well as reduce the contamination rate (Figure 2).

Biological properties-based CTC separation method usually has high purity due to the specific interaction between antigen and antibody, but its operation processes including labeling and sorting are complicated, the throughput is low (usually between 1–3 mL h^−1^), and due to a large amount of antibody is consumed, the cost is high (Table 1). More importantly, due to the EMT of CTCs during metastasis, not all cancer cells express E-type markers [37,52,53], which causes the loss and underestimation of the number of CTCs.

**Figure 1 sensors-20-06073-f001:**
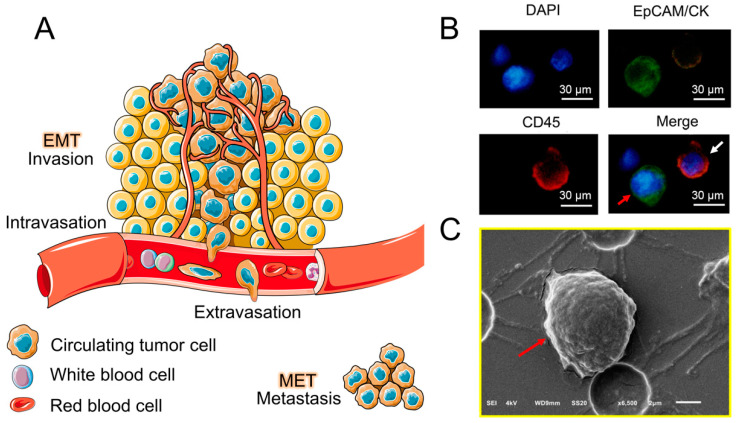
The biological properties of CTC. (**A**) A schematic diagram of the four stages of tumor metastasis formation, which is invasion, intravasation, extravasation and metastasis. (**B**) Fluorescence image of CTC (red arrow) and WBC (white arrow), cells were stained with DAPI, EpCAM/CK and CD45 (Scale bars = 30 μm) [54]. (**C**) Scanning electron micrograph of CTC (red arrow) (Scale bar = 2 μm) [55].

**Figure 2 sensors-20-06073-f002:**
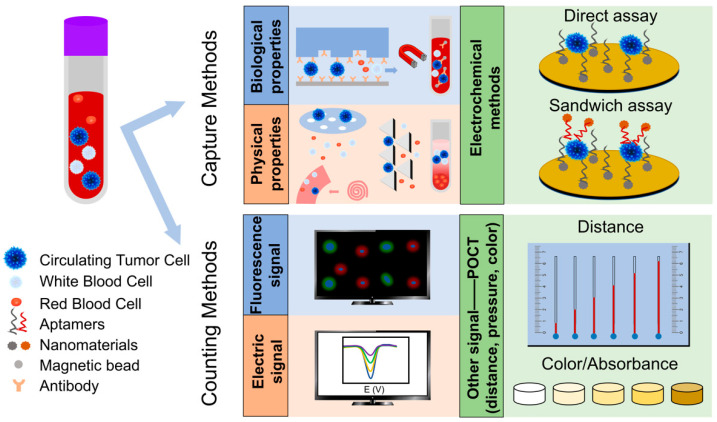
The capture methods and counting methods of CTC. The capture methods including biological properties-based methods, physical properties-based methods and electrochemical methods. The counting methods including fluorescence signal-based methods, electric signal-based methods and other signal-based methods (simple signal, such as distance, pressure and color etc.).

**Table 1 sensors-20-06073-t001:** Summary of biological and physical properties-based CTC detection methods.

Separation Principles	Technology	Flow Rate (mL h^−1^)	Effective Flow Rate (mL h^−1^)	Efficiency/Sensitivity	Capture Purity	Enrichment Factor	Clinical Sample	Ref.
**Biological properties-based methods**	**CellSearch**	3	3	80–82%	——	——	Metastatic breast, prostate or colorectal	[56]
	**CTC-chip**	1–2	1–2	>60%	49–67%	——	Non small cell lung cancer (n = 55), prostate (n = 26), pancreatic (n = 15), breast (n = 10) and colon (n = 10)	[24]
	**HB-chip**	1.2	1.2	91.8 ± 5.2%	14.0 ± 0.1%	——	Prostate (n = 15), lung (n = 4)	[25]
	**CellCollector**	30	30	10–35%	50–96.4%	——	Pancreatic (n = 43)	[57]
	**GO-chip**	1–3	1–3	84–95%	——	——	Breast (n = 10), pancreatic (n = 3)	[58]
**Physical properties-based methods**	**DLD**	600	30	85%	——	3.4	——	[59]
	**Vortex-chip**	240–360	12	20.7%	89%	3.5×10^4^	Breast (n = 4), lung (n = 8)	[60]
	**Multiplexed spiral chip**	21	21	76.4–87.6%	1CTC/30–100WBCs	~10^4^	Breast (n = 5), lung (n = 5)	[61]
	**Labyrinth-chip**	150	30	>90%	WBCs number: 13,800 mL^−1^ (once); 663 mL^−1^ (twice)	——	Breast (n = 56), pancreatic (n = 20)	[62]
	**Micro-obstacles spiral-chip**	390	22	>95%	——	2.29×10^5^	——	[63]
	**Spiral series-chip**	60–120	2.4	73.75–80.75%	63.6%	——	——	[64]
	**Multistage hydrodynamic focusing-chip**	0.54–0.90	0.90	>90%	85%	——	Ovarian (n = 1)	[65]
	**Membrane filtration**	12–60	12–60	>80%	WBCs number: 1000–3000 mL^−1^	——	——	[66]
	**Cluster-chip**	2.5	2.5	41% (two-cell)	——	——	Breast (n = 27), melanoma (n = 20), prostate (n = 13)	[26]
	**Conical-shaped holes-chip**	12	4	96%	WBC clearance efficiency: 96%	~10^2^	Lung (n = 8), nasopharynx (n=2), mediastinal (n = 2), cardia (n = 1), cervical (n = 1) and breast (n = 1)	[67]
	**Ratchets-chip**	1	1	90%	——	~10^4^	Prostate (n = 20)	[68]
	**Parsortix-chip**	10	10	54–69%	WBCs number: 200–5000 mL^−1^	——	Breast (n = 10), colon (n = 10) and lung (n = 6)	[69]

### 2.2. Methods Based on Physical Properties

The difference in physical properties can also be used to separate CTC from blood cells (including cell size, deformability, density, and dielectric properties, etc.). The most commonly used is the separation based on size and deformation. The diameter of CTC is 8–20 μm [70,71], the Young’s modulus is 494–2472 Pa [72,73,74], while the diameter of WBC is 8–15 μm [70], and its Young’s modulus is 48–156 Pa [75,76], the diameter of RBC and platelet is much smaller. Because the CTC is larger in size and is less deformable [77], it can be separated by ISET (isolation by size of epithelial tumor cells). It utilizes track-etched polycarbonate membranes with evenly distributed 8 μm-diameter cylindrical pores to filter the blood, after filtering, the larger CTCs are left on the membrane, and the smaller blood cells are filtered (Figure 2). In addition, researchers also developed various membrane filtration methods with different membrane pore sizes [66,78,79,80]. The difference in cell density can also be used to separate CTC. The sucrose density gradient centrifugation method uses sucrose to pre-form a concentration gradient, different types of cells are in different positions due to the difference in density and sedimentation coefficient, and the separation of CTC can be achieved (Figure 2) [81,82]. Taking advantage of the difference in physical properties between CTC and WBC, and by rationally designing microfluidic chips, effective separation can also be achieved. For example, by designing some pillar arrays with suitable gaps, the gap allows WBCs and RBCs to pass through, while CTCs are stuck and captured (Figure 2) [26,68,83]. This can be called filtration-based microfluidic methods. In addition to these, there are microfluidic methods based on streamlines. Deterministic lateral displacement (DLD) [54,84,85] and spiral designs [61,62,63,64,86] are often used to separate CTCs. The main principle is that cells reach an equilibrium position in the channel under the action of various forces (such as gravity, buoyancy, lateral force, and Dean force etc.), the equilibrium position is related to cell size, and then CTC can be separated from other blood cells (Figure 2).

In addition to the methods mentioned above, some researchers have also achieved size-based CTC separation by applying an external force: acoustic wave. The basic principle is that cell movement is caused by sound pressure waves generated by the transducer. The standing acoustic wave in the microfluidic channel has nodes and antinodes, cells flowing through these periodic pressure nodes and antinodes are subjected to different acoustic radiation forces, the force is proportional to the cell size, resulting in the difference in lateral displacement, and then realizes the separation of CTC [87,88]. Acoustophoresis-based CTC separation has the following advantages: it separates cells in a label-free, contact-free, and biocompatible manner, and retains cells’ original state, integrity, and function [21]. However, the problem is that the flow rate is low (1.2 mL h^−1^) [89], and the blood needs to be lysed or diluted in a large proportion. Wu et al. used a polydimethylsiloxane (PDMS)-glass hybrid channel to increase the reflection of sound waves and reduced the velocity at the center of the channel by introducing a PDMS barrier in the middle of the channel, which effectively improved the separation efficiency and increased the flow rate to 7.5 mL h^−1^ [88].

Physical properties-based CTC separation does not rely on the expression of antigen on the cell surface, and can obtain both E-type and M-type CTCs. The separation process is high throughput (usually can reach tens of mL h^−1^, which means a shorter process time), low cost and easy to operate. However, there are problems with CTCs capture efficiency and purity, because of the size overlap of WBCs and CTCs. In order to maintain high CTC capture efficiency, correspondingly, more WBCs will be captured (may reach tens of thousands per milliliter). Moreover, in order to achieve high purity, the CTC capture efficiency may decrease (Table 1).

## 3. Electrochemical Detection of CTC

In recent years, electrochemical detection of CTC has attracted widespread attention due to its advantages such as convenience, rapidity, low cost, high sensitivity, low detection limit, and strong specificity [90]. From the principle of separation, electrochemical detection of CTC is a kind of the biological properties-based CTC separation. However, it usually uses aptamer to replace traditional antibodies. Aptamer has similar specificity and affinity compared to antibodies, but it has many advantages, including easier to achieve large-scale production, lower cost, and lower immunogenicity [22,91] This greatly reduces the cost of electrochemical detection. We summarized common aptamers used for CTC detection in Table 2, and listed their Kd values, targeted biomarkers, applicable cell lines and cancer types, etc. Most of these aptamers are dsDNA, and a small part are ssDNA or RNA.

The principle of electrochemical detection is to modify the electrode using nanomaterials and cancer cell specific aptamer, and CTC is bonded to aptamer (on the electrode surface), which will cause the current/electrochemiluminescence (ECL)/photoelectrochemical (PEC) signal to decrease or EC impedance spectroscopy (EIS) signal to increase [27,92,93,94,95,96]. By establishing a linear relationship between the change and the number of CTC, the quantification can be realized. The electrochemical CTC detection methods can be roughly divided into direct assays, sandwich assays and other assays [22]. The direct assays use nanomaterials-aptamer as the capture probe (which is directly modified on the electrode) to bind and detect CTC (Figure 2) [97]. The sandwich assays add a signal probe on the basis of direct assays, that is, after CTC is bound to the capture probe on the electrode, a signal probe (another kind of nanomaterials-aptamer) is added. After the signal probe is combined with CTC, it can significantly amplify the signal, thereby increasing the sensitivity and reducing the detection limit (Figure 2) [30,98,99]. We summarized the electrochemical-based CTC detection methods in Table 3, including the direct assay and the sandwich assay, and we listed the assay time, linear range, limit of detection, capture probe, and signal probe.

**Table 2 sensors-20-06073-t002:** Summary of aptamers used for CTC detection.

Cancer Type	Cell Line	Aptamer	DNA or RNA	Biomarker	Kd/nM	Ref.
Breast	MCF-7/HeLa	AS1411	DNA	Overexpressed nucleolin on the cell surface	100–1000	[100]
	MCF-7	MUC 1	DNA	Overexpressed mucin 1 (MUC1) glycoprotein on the cell membrane	38.3	[101]
	MCF-7/MDA-MB-231	SYL3C	DNA	EpCAM	38 ± 9	[102,103]
Leukemia	CCRF-CEM	Sgc8c	DNA	Tyrosine kinase-7 on the cell surface	2.0 ± 0.2	[104]
Lymphoma	Toledo	Sgd5	DNA	——	——	[105]
	Ramos	Td05	DNA	B-cell receptor	74.8 ± 8.7	[105,106]
	Ramos	TE02	DNA	——	0.76 ± 0.13	[106]
	Karpas 299	Anti-CD30 RNA aptamer	RNA	CD30	——	[107]
Colorectal, glioblastoma and lung	——	Anti-EGFR RNA aptamer	RNA	Epidermal growth factor receptor (EGFR)	2.4	[108,109]
Liver	Mear	TLS1c	ssDNA	——	9.79 ± 0.30	[110,111]
	HepG2/Mear	TLS11a	DNA	——	4.51 ± 0.39	[110,111]
Lung	——	LC-18	DNA	Neutrophil defensin 1 and 3	38	[112]
Prostate	LNCaP	A10 RNA aptamer	RNA	Prostate specific membrane antigen (PSMA)	600	[113,114]

### 3.1. Direct Assays

The principle of direct assays is based on the combination of CTC and electrode, thus affects the transfer of electrons, which in turn leads to a decrease in current/increase in electrical impedance [94,115], thereby realizing the quantitative detection of CTC. Many researchers use functionalized nanomaterials to modify the electrodes, the main purpose is to increase the contact area, enhance the conductivity, and then enhance the signal. Cao et al. proposed a method for detecting acute leukemia CCRF-CEM cells using a hybrid of nanochannels and ion channels arrays. The main principle is after modifying the sgc8c aptamer on the surface of the channel, CTC is combined with ion channel, then it will dramatically block the ionic flow through channels, which in turn causes current changes, using linear sweep voltammetry (LSV) cytosensor to record the change and to quantify CTC. Compared with a single channel, the array channels could amplify the signal and to improve the detection sensitivity. The linear range is 1 × 10^2^–2 × 10^6^ cells mL^−1^, and the limit of detection (LOD) is 100 cells mL^−1^ (Figure 3A) [116]. Subsequently, Li et al. reported a PEC-based CTC detection method, hypotoxic ternary mercaptopropionic acid (MPA)-capped AgInS_2_ nanoparticles (NPs) as PEC sensing substrates, after being excited by red light, it shows high photon-current conversion efficiency, and generates strong photocurrent. After CCRF-CEM cell is connected to aptamers on AgInS_2_ NPs, the photocurrent is significantly reduced, and CTC can be detected. Its linear range is from 1.5 × 10^2^ to 3.0 × 10^5^ cells mL^−1^, and the LOD is 16 cells mL^−1^ (Figure 3C) [93]. In order to further improve the detection sensitivity, Wang et al. prepared nanomaterials-plasmonic gold nanostars (AuNSs), modified sgc8c aptamer on its surface and fixed it on glassy carbon (GC) electrode. The results showed that the LOD for CCRF-CEM cell is as low as 5 cells mL^−1^, in addition, MCF-7 cells could also be detected with a LOD of 10 cells mL^−1^ (Figure 3B) [117]. Liu et al. used ECL cytosensor for MCF-7 cell detection. Modified Au@CDs-aptamer on GC electrode, after adding K_2_S_2_O_8_, Au@CDs showed strong electrochemiluminescence. Due to the combination of CTC, the strength of ECL will decrease. Its linear range was 100 to 10,000 cells mL^−1^ with a LOD of 34 cells mL^−1^ (Figure 3D) [92]. Graphene nanomaterials are also widely used in electrochemical detection of CTC due to their excellent electrical conductivity [118]. For example, Zhang et al. synthetized a xFe_2_O_3_-nPt (or nPt-xFe_2_O_3_)-coated graphene nanostructure (xFe_2_O_3_-nPt@graphene) to detect MCF-7 cells using differential pulse voltammetry (DPV) cytosensor. The combination of graphene, Pt and Fe_2_O_3_ greatly reduced the electrode resistance and improved the electron transfer efficiency, and effectively amplified the signal and improved the sensitivity. The linear range is 18 to 1.5 × 10^6^ cells mL^−1^, and the LOD is 6 cells mL^−1^ [119]. Bábelová et al. invented a method to detect leukemia, which is based on electrochemical impedance spectroscopy (EIS) and thickness shear mode acoustic method (TSM). It used sgc8c as the specific aptamer and achieved high-sensitivity detection of Jurkat cell. The LOD of electrochemical and acoustic sensors is 105 ± 10 and 463 ± 50 cells mL^−1^, respectively [120].

**Table 3 sensors-20-06073-t003:** Summary of electrochemical CTC detection methods.

	Assay Time (min)	Linear Range (Cells mL^−1^)	LOD (Cells mL^−1^)	Aptamer/Antibody	Target Cell	Capture Probe	Signal Probe	Clinical Sample	Ref.
**Direct assay**	70	18–1.5 × 10^6^	6	AS1411	MCF-7	xFe_2_O_3_-nPt (or nPt-xFe_2_O_3_)-coated graphene nanostructure+aptamer	——	Spiked blood samples	[119]
	45	1 × 10^2^–2 × 10^6^	100	sgc8c	CCRF-CEM	Nanochannel−ion channel +aptamer	——	——	[116]
	90	100–10,000	34	anti-MUC 1 aptamer	MCF-7	Gold electrode+Au@CDs +aptamer	——	Human serum samples	[92]
	45	5–1 × 10^5^	5	sgc8c	CCRF-CEM	Glassy carbon electrode +AuNSs+aptamer	——	Human serum samples and spiked blood samples	[117]
	60	1.5 × 10^2^–3 × 10^5^	16	sgc8c	CCRF-CEM	AgInS_2_ NPs-modified electrode+aptamer	——	——	[93]
**Sandwich assay**	120	75–5500	75	E-cadherin	Not limited	CNT-AuNPs+antibody	E-cadherin antibody+QD	Fetal calf serum and mouse blood	[121]
	50	1 × 10^2^–5 × 10^4^	5	SYL3C	MCF-7	m-aptamer/MCH/AuE	Anti-EpCAM/HRP-AuNP	Spiked blood samples	[28]
	20	5–500	4	Td05	Ramos	AuNPs-Fe_3_O_4_-GS+aptamer	AuNPs with the electroactive species (Fc-SH/Thi) +aptamer	Spiked blood samples and 3 leukemia patients	[98]
			3	Sgc8	CCRF-CEM				
	40	10–1 × 10^6^	2	EpCAM+vimentin	MCF-7	Ketjen black+AuNP +antibody	PdIrBPMNS+antibody	Spiked blood samples	[29]
	170	3–3000	1	5′-thiol modified MCF-7 binding aptamer	MCF-7	MN+EpCAM	Cu_2_O+aptamer	Spiked blood samples	[30]
	60	3–1000	1	anti-MUC 1 aptamer	MCF-7	MN+EpCAM	LiFePO_4_/Au+aptamer	Spiked blood samples	[27]
	115	5–3 × 10^4^	1	SYL3C-RCA primer	MCF-7	MN+EpCAM	Aptamer+DNA amplification (RCA)	Spiked blood samples	[99]
	40	20–1 × 10^6^	3	EpCAM	MCF-7	Au electrode+AB +AuNP+proteinG+antibody	Pt@AgNFs+antibody	Spiked blood samples	[94]

### 3.2. Sandwich Assays

The sandwich assays add a signal probe on the basis of direct assays. The capture probe is fixed on the electrodes to identify and capture CTC. The signal probe is used for amplifying the signal, improving the sensitivity, and reducing the detection limit [27,94,122]. Signal probes are usually some active nanomaterials or particles with enzymatic activity.

Shen et al. used magnetic nanospheres modified with anti-EpCAM antibody to capture MCF-7 cells. The captured MCF-7 cells are spread on the electrode surface, aptamer (SYL3C-RCA primer) is added and bound to the cells, by rolling circle amplification (RCA), producing a large amount of DNA molecules and reacting with the substrate to amplify the electrochemical current signal. The linear range is 5–3 × 10^4^ mL^−1^, and the detection limit can reach 1 cell mL^−1^ (Figure 4F) [99]. RCA technology was also used by other groups [28,123]. For example, Yang et al. used RCA reaction to expand the initial PD/CDT so that the expanded DNA chain contains multiple duplicate aptamers. The expanded DNA strand as a capture probe has the following advantages: (1) This flexible long DNA chain can extend to the depth of the cell suspension, which allows the aptamers in full contact with the target cells; (2) The aptamer binds to the target cell in a multivalent manner, effectively improve the capture efficiency of target cells. The signal probe anti-EpCAM/Horseradish peroxidase (HRP)-gold nanoparticles (AuNP) is used to catalyze H_2_O_2_ to generate electrical signals for detection. The linear range is 1 × 10^2^–5 × 10^4^ mL^−1^, and the LOD is 5 cell mL^−1^ (Figure 4E) [28].

Tang et al. modified AuNPs/Acetylene black (AuNPs/AB) on the gold electrode to increase the specific surface area and enhance the conductivity of the gold electrode. Then anchored antibody on the surface of AuNPs/AB to capture MCF-7 cells. After capture, the signal probe Pt@AgNFs was added, which can catalyze H_2_O_2_ to generate electrical signals. The linear range is 20–1 × 10^6^, and the detection limit is 3 cells mL^−1^ (Figure 4A) [94]. Luo et al. proposed a PEC biosensor-based CTC detection method with higher sensitivity using antibody-magnetic nanospheres to capture MCF-7 cells. After spreading the capture cells on the electrode, the photocurrent intensity of hexagonal carbon nitride tubes (HCNT) is reduced due to the steric hindrance derived from MCF-7. After adding a signal probe Cu_2_O NPs, the photocurrent intensity of was further decreased because Cu_2_O NPs competitively absorbed the excitation light, and the aptamer molecules further increased the steric hindrance, the linear range is 3–3000 cell mL^−1^, and the detection limit is 1 cell mL^−1^ (Figure 4D) [30]. Similarly, Zhang et al. also used antibody-magnetic nanospheres to capture MCF-7 cells, and after coating the cells on the electrode, used gold nanoparticle-modified LiFePO_4_ coupled with the aptamer as a signal probe, due to the reaction of phosphate group in LiFePO_4_ with molybdate that formed redox molybdophosphate (PMo_12_O_40_) precipitates and caused current change. The linear range is 3–1000 cells mL^−1^, and the LOD is 1 cell mL^−1^ (Figure 4B) [27]. The above parts are all about the detection of one type of cell. Using multiple different aptamers can simultaneously detect multiple types of cells. For example, Dou et al. used AuNPs-Fe_3_O_4_-graphene nanosheets (GS) as the capture probe, AuNPs coupled with aptamer (Td05/Sgc8) and decorated with the electroactive species (Fc-SH/Thi) as the signal probe, and simultaneously achieved the quantitative detection of Ramos and CCRF-CEM cells, the detection limit is 4 and 3 cells mL^−1^, respectively. Further, the linear range is 5–500 cells mL^−1^. More importantly, they also performed testing of three leukemia patients and showed obvious results (Figure 4C) [98]. Du et al. proposed a Cyclic voltammetry (CV) cytosensor based CTC electrochemical detection, and it can specifically detect the change of E-cadherin and analyze different stages of EMT. They modified CNT-AuNPs on the electrode surface, and then used quantum dot (QD) coupled anti E-cadherin antibody to bind to CTC. QD itself acts as a signal probe to enhance electrical signal, and at the same time, QD can also provide a strong fluorescent signal for imaging [121].

In Table 3, we can see that, compared to the direct method, sandwich assays have higher sensitivity, and its LOD can be as low as 1 cell mL^−1^. However, the detection time may be longer. In addition, the fabrication and detection process are also more complicated. The common problems are the narrow range of applicable cell types and very few clinical trials.

### 3.3. Other Assays

There are some electrochemical methods that only use electrochemical principles to achieve the enrichment and capture (or release) of CTC, while they do not use photoelectric signals to quantify CTC. We classify these methods as other assays [124,125].

Yan et al. set a series of elliptical pillars on the surface of the Polydimethylsiloxane (PDMS) chip, coated with electroconductive gold layer, and then coupled with EpCAM antibody to capture CTC. A capture efficiency of 85−100% was achieved for different cell lines both in buffer and in blood. More importantly, because the chip is conductive, it can easily release the captured cells for subsequent culture (–1.2 V, 10 min, survival rate after release >95%) and lysis the cells for PCR analysis (20 V, 10 min). It is worth mentioning that this method also has a high capture efficiency for cells with low EpCAM expression (Figure 5C) [125]. Similarly, Zhai et al. also reported a highly efficient method to capture and release CCRF-CEM cells, but using aptamers modified gold nanowire arrays (AuNWs). The distance between AuNWs is narrower, about 110 to 130 nm. Compared with directly coupling aptamer on the flat gold substrate, aptamers modified AuNWs effectively improved the capture efficiency of CCRF-CEM Cells (from 9% to 83%). CCRF-CEM Cells could be easily released by applying a voltage of 1.2 V for 30 s, the release efficiency is 96.2%, and the survival rate after release is about 90% (Figure 5B) [126]. Subsequently, Zhang et al. modified APBA+dopamine+sgc8c on the surface of poly(APBA-co-ABSA), and the capture efficiency of CCRF-CEM cells reached 83%. After applying 0.6 V voltage for 120 s, the boronic ester linkage between boric acid and dopamine was broken, and 88% of the cells were released, and the cell survival rate after release was 98%. This method provides a gentle, fast, and non-invasive release of cells (Figure 5D) [127]. Gurudatt et al. designed an electrochemical-based CTC separation and enrichment method. By modifying the lipid bilayer on the surface of the channel and applied an AC electric field to make the cells move in a wave-like manner in the channel, and the passage time is related to the cell size and charge. Therefore, the separation of CTCs from WBCs and RBCs is realized. The separate efficiency is 92.0 ± 0.5%. The detection of CTC is based on the redox reaction of DM (daunomycin) molecules (coated on the CTC) to generate electrical signals. Blood samples from 37 cancer patients were tested with 90.9% detection rate (Figure 5A) [128].

## 4. Detection Method Based on Point-of-Care Testing (POCT)

POCT refers to a clinical test that can be performed on the patient’s bedside without the need for complicated equipment and specialized laboratory examiners. It samples on-site and analyzes immediately, eliminating the complicated processing procedures required for sample testing in the laboratory and quickly obtaining medical test results [129]. At present, the most commonly used POCT include blood glucose test paper and urine pregnancy test paper. In addition, POCT has other applications, including serum separation, detection of enzyme activity, detection of bacteria, viruses, proteins, DNA, etc. [31]. The methods based on POCT can be divided into self-driving testing and external force-driving testing according to driving force [31]. Self-driving detection includes the use of capillary effect, negative pressure, or chemical reaction to generate gas [32,130], etc. to promote the reaction and output related signals, the external force-driving detection usually uses hand push, finger trigger button etc. as the driving force to promote the reaction. The most important thing is that the output signal of POCT is very simple and easy to get, usually including color, distance, pressure, temperature, etc. [131,132,133,134,135].

### 4.1. POCT Based Detection of Enzyme Activity/Proteins/Compound

Wang et al. immobilized telomerase primer (TS) on the surface of magnetic bead (MB), and the primer could be extended under the action of telomerase, and formed a single-stranded DNA containing (TTAGGG) repeating unit, the repeating unit can specifically bind to cDNA- nanoparticles (PtNPs), using the H_2_O_2_ catalytic ability of PtNPs, adding substrate H_2_O_2_, the gas is generated and the pressure changed. Using a portable pressure gauge to detect the change of the pressure could achieve the detection of single cell telomerase activity (Figure 6C) [134]. Ma et al. also designed a POCT based method to detect telomerase activity. They hybridized the magnetic beads with the capture DNA and the report DNA-functionalized enzyme. In the presence of telomerase, the TS primer was extended to generate a repetitive sequence and matched with the report DNA-functionalized enzyme. As the concentration of telomerase increased, more reported DNA-functionalized enzyme (includes alkaline phosphatase (ALP) and galactosidase (Gala)) remained in the solution, and Gala catalyzed luciferin di-β-D-pyridine Galactoside (FDG) hydrolyzed into FMG to produce a fluorescent signal. ALP catalyzes p-nitrophenyl phosphate (pNPP) to produce dark yellow p-nitrophenol (pNP), which caused a color change for colorimetric determination. In addition, ALP also catalyzed the hydrolysis of ATP to AMP, which can be detected by a portable ATP meter. The detection limit can be as low as a single cell mL^−1^, and it has been verified by patients [136]. Cao et al. quantified ALP activity by measuring distance. The entire system only needs an electrophoresis titration (ET) chip, a lithium battery, a UV LED, and an iPhone as a recorder, without other complicated power supplies and fluorescence detectors. The principle of the testing is based on the ability of ALP to react with the substrate. The product can produce fluorescence under UV excitation, and could move under an external electric field, and interacted with Tris-HCl buffer to produce a moving reaction boundary (MRB). ALP activity could be quantified by measuring the distance of MRB. The mobile phone can be used for data collection and analysis (Figure 6B) [132]. Recently, Wang et al. also used a smart phone, to determine anthrax based on color change, using Eu(III) functionalized carbon dots (CDs-Eu) as a fluorescent probe, where DPA (an anthrax biomarker) can interact with CDs-Eu, resulting in a color change. Using the color analysis software on the smartphone, DPA can be quantified, and its linear range is 0.5 nM to 5 μM with a LOD of 0.8 nM (Figure 6D) [131]. Xiao et al. designed an optical microfiber sensor modified by a polystyrene@gold nanosphere to detect the (CEA)-related cell adhesion molecules 5 (CEACAM5) concentration in serum. After CEACAM5 was bonded on the microfiber surface, the refractive index (RI) increased and the transmission of the optical sensor was found to redshift. The distance of the redshift is related to the concentration of CEACAM5. The LOD is 3.54 × 10^−17^ M in pure solution, 5.27 × 10^−16^ M in serum, about 6 orders of magnitude lower than current methods [137]. The proposed medical smartphone-powered dongle was demonstrated to be a very promising platform as a miniaturized electrochemical analyzer for point-of-care monitoring of the critical biochemical parameters such as blood ketone and a good solution for mobile health management. Ainla et al. designed a “universal wireless electrochemical detector” (UWED). It can perform routine electrochemical detection such as potentiometry, chronoamperometry, cyclic voltammetry and square wave voltammetry, at the same time, it has the following advantages compared with benchtop commercial potentiostats, which is simple, small in size, and inexpensive. This detector was connected to a smartphone (or a tablet) using “Bluetooth Low Energy” protocol, and the results of electrochemical detection can be immediately displayed on the mobile phone. It facilitates POCT-based electrochemical detection [138]. Guo et al. also proposed a POCT-based electrochemical method, they used dongle which is similar to UWED mentioned above. Their main purpose is to detect blood ketone in finger whole blood, it can help us to assess the progress of diabetic ketoacidosis (DK) and diabetic ketosis acid (DKA) diabetes. To be specific, they connected a mobile phone, a dongle, and a disposable electrochemical ketone test strip together. Among them, the test strip as a biosensor, underwent a series of redox reactions to convert the concentration of ketone into electric current. The dongle acted as a miniaturized electrochemical analyzer, and helped to present electrochemical results on the mobile phone. The whole system was cheap and simple, and its LOD can reach 0.001 mmol L^-1^ with a linear range of 0.001–6.100 mmol L^−1^ [139].

### 4.2. POCT Based Detection of CTC

The above tests are all about detecting some simple enzyme activities or simple proteins/compounds. Due to the catalytic activity and high specificity of enzymes, the definite structure of protein or compounds, it is easy to find suitable substrates or suitable signal probes to detect them. However, it is difficult to detect CTC by POCT, because the number of CTC in the blood is rare, and there are a large number of background cells (including WBCs and RBCs) in the blood. These background cells can interfere with the detection of CTC. This poses a challenge to the sensitivity and specificity of the detection system. Apart from this, the heterogeneity of CTC makes its surface composition very complex [15,140,141], and it is difficult for us to directly quantify using a simple general method, since there is a lack of CTC markers that are truly universally applicable with highly specificity. Recently, Abate et al. labeled CTC with aptamer-conjugated nanoparticles (ACNPs), and ACNP can catalyze the decomposition of H_2_O_2_. The by-product (oxygen) of the catalytic reaction caused the red ink to move, the moving distance is linearly related to cell concentration. The detection limit can be as low as a single cell mL^−1^, and the linear range is 0–2000 cells mL^−1^ (Figure 6A) [133]. Moreover, the change of temperature could also be used to detect cancer cells. Zhang et al. connected graphene oxide (GO) and anti-EpCAM antibodies to microbeads, and microbeads could bind to the surface of cancer cells. Since magnetic microbeads and GOs have the photothermal effect, they will generate temperature changes when irradiated with laser. By establishing a linear relationship between temperature changes and the number of cancer cells, quantitative detection of cancer cells can be achieved, the detection limit is 100 cells mL^−1^ [135]. Yang et al. used two kinds of antibodies as capture probe and signal probe, respectively, and realized POCT-based CTC detection. The specific process is to use SK-BR-3 cells to simulate CTCs in blood, use anti-EpCAM antibody-functionalized magnetic beads as the capture probe to enrich and separate tumor cells, and then use anti-HER2 antibody and invertase co-modified polystyrene microspheres as signal probes. The substrate sucrose is converted into glucose under the action of invertase. By using the common glucose meter transducer, CTC detection was achieved. The linear range is 50–1000 cells mL^−1^ with a detection limit of 7 cells mL^−1^ [142]. Xia et al. came up with an electrochemical-based point-of-care CTC detection method. CTCs were firstly captured by aptamer-modified magnetic beads, and then they can sequestrate ferroceneboronic acid (FcBA) or 4-mercaptophenylboronic acid (MPBA) by the formation of boronate ester bonds, thus leading to the decrease in the electrochemical signal of FcBA or preventing the MPBA-triggered aggregation of AuNPs, which leads to changes in electrochemical signal or color. The colorimetric assays achieved a linear range of 50–1.5 × 10^4^ cells mL^−1^ with a sensitivity of 50 cells mL^−1^ [143].

## 5. Conclusions and Future Trends

In this review, we summarized the latest developments in electrochemical detection of CTC. In general, the detection of CTC by electrochemical methods has the following advantages: convenient, fast, low cost, high sensitivity and does not require complex equipment. However, there are also some problems which limit its clinical application: (1) The types of stable and high-affinity aptamers are too limited, only a few types of cancer cells could be used, mainly including CCRF-CEM cells and MCF-7 cells. (2) The sensitivity is still not high enough, it is difficult to achieve a LOD of 1 mL^−1^. (3) The current tests are more biased towards the laboratory, and the real patient-based blood tests are few. Compared to the medium system, the capture efficiency of blood-based tests will decrease, mainly due to the interference of ultra-high density of blood cells and complex components in the blood [99], which will cover the capture site or affect the progress of the reaction, thereby affecting the output of the electrochemical signal. For the popularization of the clinical application with electrochemical detection of CTC, efforts need to be made in the following directions: (1) Find more suitable CTC universal aptamers, because aptamer determines the specificity and accuracy of the detection, and determines the types of cancer that can be detected. More importantly, CTC are usually different from laboratory cell lines, the application of aptamers on CTC needs further exploration; (2) Further improve the sensitivity of detection, develop new catalytic nanomaterials as capture probes/signal probes to further amplify electrochemical signals; (3) Try to avoid interference caused by non-specific adsorption of blood cells, which may require us to pre-enrich the CTC in the blood. By integrating pre-enrichment strategy and electrochemical-based detection may be a more efficient and convenient method.

As for the application of POCT in CTC detection, although there are not many studies reported so far, it is a very promising development trend. Because clinical CTC detection is extremely important, and POCT provides a reliable, simple, portable, fast, and low-cost method, we proposed possible implementation methods for POCT-based CTC detection in the following section. Firstly, integrating the CTC capture methods with POCT-based detection. Using CTC capture methods to separate and enrich the CTC in advance, including magnetic separation of CTC by antigen-based method, or enrichment of CTC by physical properties-based method, or capture and release of CTC by electrochemical method. And then introduce the enriched CTC into small chambers, add aptamer/antibody-microbeads (with catalytic properties) and add substrate, the microbeads can catalyze the reaction, induce color change, or generate gas, and by recording the color change/pressure change/pressure-driven distance, achieve the quantification of CTC. The key step is to select CTC specific aptamers, select microbeads with suitable enzyme catalytic activity and select suitable reaction substrates, which can eventually generate easy-to-detect signals. Secondly, simplify the separation and enrichment method as much as possible. For example, find ways to make magnetic separation smaller and easier to carry, use hand push instead of push pump, miniaturize the electrode and power supply. Finally, it is a very wise idea to combine smart phones with POCT, because mobile phones, as a portable tool, facilitate people to collect, process, and present results [144]. At present, people have tried various methods to use measurement function, camera function, and color analysis capabilities of the mobile phone, and have also developed various software to make the mobile phone suitable for POCT detection. Many studies have shown that bright-field detection, colorimetric detection, fluorescence detection, and electrochemical detection can be realized on mobile phones. The test samples also cover blood, tears, sweat, saliva, and urine, etc. For example, Knowlton et al. used a magnetic field to separate cells, and then used the camera of the mobile phone matched with appropriate optical components to perform brightfield and fluorescence imaging of the cells. The entire system is very small and does not require large magnetic separation device and fluorescence microscope, and the separation and quantification can be achieved by only a mobile phone and a few simple components [145].

Although the current electrochemical detection of CTC is less used in clinical practice, we believe that electrochemical method, as an effective and convenient method, has developed rapidly in recent years and can be used in clinical practice in the near future. At the same time, we believe that achieving POCT-based CTC detection through suitable integration is important. This is the development trend of CTC clinical testing and the future direction of testing.

## Figures and Tables

**Figure 3 sensors-20-06073-f003:**
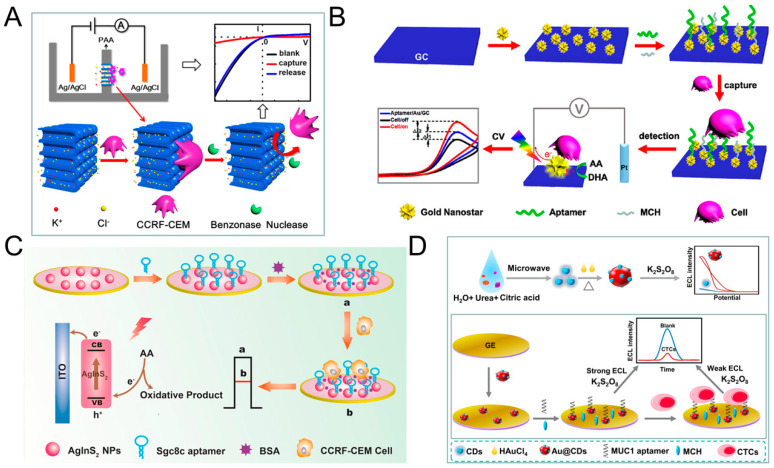
Direct electrochemical detection of CTC. (**A**) Linear sweep voltammetry (LSV) cytosensor for CCRF-CEM cell detection, using hybrid nanochannels and ion channels. Reprinted with permission from ref [116]. Copyright 2017, American Chemical Society. (**B**) Cyclic voltammetry (CV) cytosensor for CCRF-CEM cell and MCF-7 cell detection, by modifying plasmonic gold nanostars (AuNSs)-aptamer on glassy carbon (GC) electrodes. Reprinted with permission from ref [117]. Copyright 2019, American Chemical Society. (**C**) Photoelectrochemical (PEC) cytosensor for CCRF-CEM cell detection, using AgInS_2_ nanoparticles (NPs) coupled with sgc8c aptamer which exhibited high photon-to-current conversion efficiency under red light excitation. Reprinted with permission from ref [93]. Copyright 2019, Elsevier. (**D**) Electrochemiluminescence (ECL) cytosensor for MCF-7 cell detection using Au@CDs-aptamer, after adding K_2_S_2_O_8_, Au@CDs showed strong electrochemiluminescence. Reprinted with permission from ref [92]. Copyright 2020, Elsevier.

**Figure 4 sensors-20-06073-f004:**
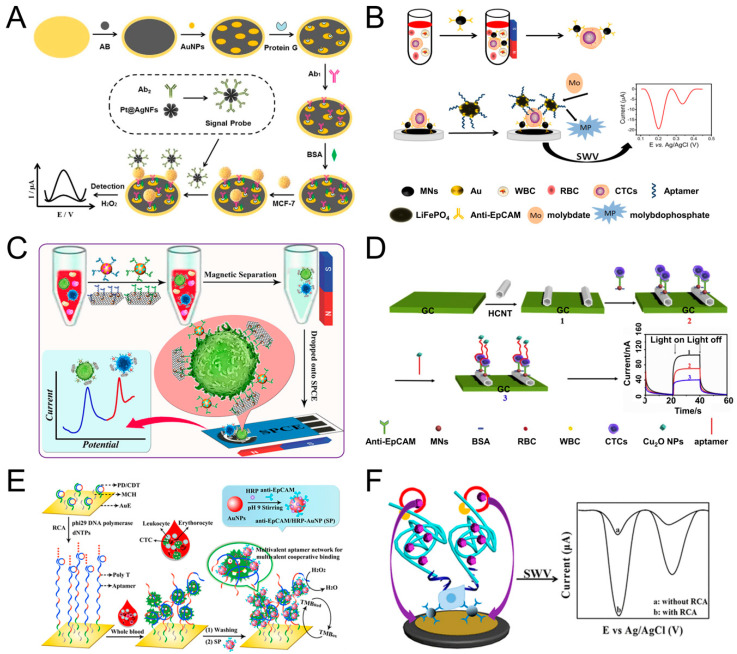
Sandwich electrochemical detection of CTC. (**A**) Voltammetric cytosensor for differential pulse voltammetry (DPV) detection of MCF-7 cells, using AuNPs/Acetylene black-antibody as capture probe and Pt@Ag nanoflowers-antibody as signal probe. Reprinted with permission from ref [94]. Copyright 2018, Elsevier. (**B**) Voltammetric cytosensor for square wave voltammetry (SWV) detection of MCF-7 cells, anti-EpCAM antibody-modified Fe_3_O_4_ magnetic nanospheres (MNs) were used to capture CTC, and gold nanoparticles modified LiFePO_4_ (LiFePO_4_/Au) particles were used as signal probe. Reprinted with permission from ref [27]. Copyright 2020, Elsevier. (**C**) Voltammetric cytosensor for SWV detection of CCRF-CEM cells, Au nanoparticles (AuNP) array-decorated magnetic graphene nanosheet as capture probe and aptamer/electroactive species-loaded AuNP as signal probe. Reprinted with permission from ref [98]. Copyright 2019, American Chemical Society. (**D**) Photoelectrochemical (PEC) cytosensor for detection of MCF-7 cells, Magnetic Fe_3_O_4_ nanospheres (MNs) as capture probe and Cu_2_O nanoparticles as signal probe. Reprinted with permission from ref [30], Copyright 2020, Elsevier. (**E**) Cyclic voltammetry (CV) cytosensor for MCF-7 cell detection, via rolling circle amplification (RCA) extension of the electrode-immobilized primer/circular DNA complexes as the capture probe, and anti-EpCAM/HRP-AuNP as signal probe. Reprinted with permission from ref [28]. Copyright 2020, American Chemical Society. (**F**) Voltammetric cytosensor for SWV detection of MCF-7 cells, anti-EpCAM antibody-modified magnetic nanospheres were used to capture and enrich CTC, and aptamer−primer DNA sequence as a signal probe, by RCA, produced a large amount of DNA molecules and reacted with the substrate to amplify the signal. Reprinted with permission from ref [99]. Copyright 2019, American Chemical Society.

**Figure 5 sensors-20-06073-f005:**
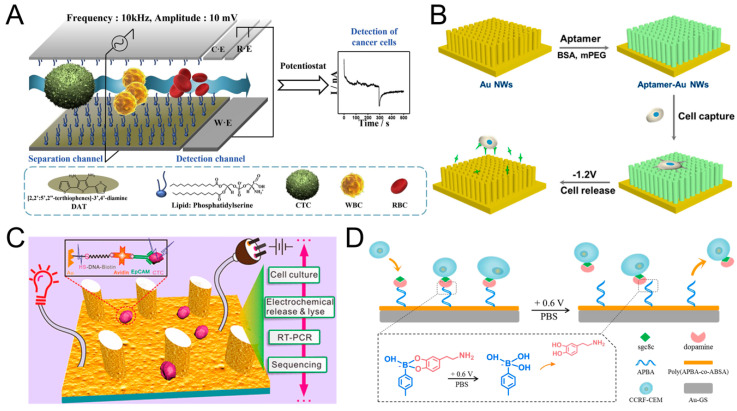
Other electrochemical detection methods of CTC. (**A**) Modified the surface of the channel with lipid bilayer, and realized CTC separation and detection by applying AC electric field. Reprinted with permission from ref [128]. Copyright 2019, Elsevier. (**B**) Fabricated gold nanowires coupled with aptamer, and achieved CTC capture and release. Reprinted with permission from ref [126]. Copyright 2017, American Chemical Society. (**C**) Set a series of elliptical columns on the surface of the PDMS chip, coated with Au, and realized the capture release and lysis of CTC. Reprinted with permission from ref [125]. Copyright 2017, American Chemical Society. (**D**) Modified APBA-dopamine-sgc8c on the surface of poly(APBA-co-ABSA), promoted the capture and fast, efficient release of CTC. Reprinted with permission from ref [127]. Copyright 2019, Elsevier.

**Figure 6 sensors-20-06073-f006:**
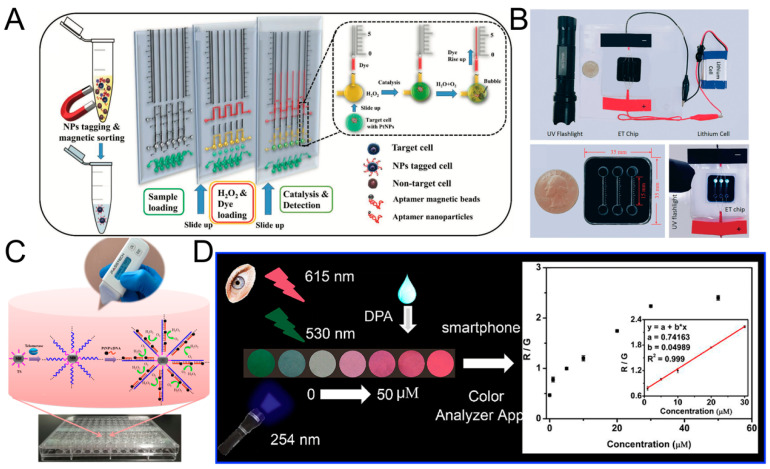
Point-of-Care testing devices. (**A**) Detecting the number of CTC based on distance, by labelling CTC with aptamer-conjugated nanoparticles (ACNP), which has hydrogen peroxide catalytic activity, in the presence of hydrogen peroxide, gas is generated to push the red ink to move, and the distance can be read. Reprinted with permission from ref [133]. Copyright 2019, John Wiley and Sons. (**B**) Quantify the activity of alkaline phosphatase (ALP) by distance. Reprinted with permission from ref [132]. Copyright 2018, Royal Society of Chemistry. (**C**) Use a portable pressure gauge to detect the pressure, to detect the telomerase activity of a single cells. Reprinted with permission from ref [134]. Copyright 2017, American Chemical Society. (**D**) A color-based method to detect anthrax biomarker DPA. Reprinted with permission from ref [131]. Copyright 2020, Elsevier.

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
