# Peer review of "Electrochemical Detection and Point-of-Care Testing for Circulating Tumor Cells: Current Techniques and Future Potentials"

_sensors, 2020, doi:10.3390/s20216073_

Round 1
Reviewer 1 Report
Authors presented a review about CTC test. The sensing principles and application examples authors introduced was clear to give readers a big picture of the CTC test. I would like to recommend public of the review after some revisions.
- Authors gave two methods separating CTC from blood. But, authors didn’t give some number that readers can use to evaluate separating efficiency of the different methods. For example, authors declared ‘there are problems with CTCs capture efficiency and purity, because of the size overlap of WBCs and CTCs.’ How about capture efficiency difference between biological properties-based and physical properties-based methods
- Authors introduced electrochemical detection of CTC as an important part of the review. As authors said, aptamer worked as sensitive components of the electrochemical detections. Is there electrochemical detection still using antibody as sensitive components?
- Following the above question, if there are electrochemical detection using antibody as sensitive components, what is difference between the electrochemical detection using antibody and aptamer.
- In section 3.1, a PEC-based CTC detection reported by Li et al. has a linear range from 1.5×102 to 3.0×105 cells mL-1. But, authors declared that amount of CTC in the blood is about 1-100 cells mL-1. Does sensing range of the PEC-based detection match clinic application of CTC detection?
- In Point-of-Care Testing section, all methods seem optical. Are there some examples employing electrochemical detections for POCT applications?
- Texts in Figure 3 are too small. I can’t see texts of panel D even if 200% magnified.
Author Response
Our response could also be found in the attachment.
Referee: 1
Comments:
Authors presented a review about CTC test. The sensing principles and application examples authors introduced was clear to give readers a big picture of the CTC test. I would like to recommend public of the review after some revisions.
Our response: We thank the reviewer for the positive assessment.
1. Authors gave two methods separating CTC from blood. But, authors didn’t give some number that readers can use to evaluate separating efficiency of the different methods. For example, authors declared ‘there are problems with CTCs capture efficiency and purity, because of the size overlap of WBCs and CTCs.’ How about capture efficiency difference between biological properties-based and physical properties-based methods
Response & Revisions: Thanks for the reviewer’s comments and suggestions. We added a new Table 1 (on p.5), summarized the biological and physical properties-based CTC detection methods in detail, listed the specific capture efficiency, flow rate and purity, etc. Thanks.
2. Authors introduced electrochemical detection of CTC as an important part of the review. As authors said, aptamer worked as sensitive components of the electrochemical detections. Is there electrochemical detection still using antibody as sensitive components?
Response: As mentioned in the article, electrochemical-based CTC detection usually uses aptamer as the sensitive component, but in some cases, it also uses antibodies. For example, there are some methods that still use anti-EpCAM or anti-vimentin antibodies (Table 3 on p.9).
3. Following the above question, if there are electrochemical detection using antibody as sensitive components, what is difference between the electrochemical detection using antibody and aptamer.
Response: Aptamer has similar specificity and affinity compared to antibodies, and the final achieved sensitivity are similar. But it has many advantages, including easier to synthesize and modify, can achieve large-scale production, lower cost, and lower immunogenicity.
4. In section 3.1, a PEC-based CTC detection reported by Li et al. has a linear range from 1.5×102 to 3.0×105 cells mL-1. But, authors declared that amount of CTC in the blood is about 1-100 cells mL-1. Does sensing range of the PEC-based detection match clinic application of CTC detection?
Response: Li et al reported a PEC-based CTC detection method, its linear range is from 1.5×102 to 3.0×105 cells mL-1 with a limit of detection of 60 cells mL-1. If the number of CTCs is 0-60 cells mL-1, it couldn’t be detected by this method. If the number of CTCs is 60-100 cells mL-1, the change of signal can be detected, but the quantification of the number of CTCs cannot be achieved. Therefore, we could say that it is not suitable for clinical application. This is also the reason why people are committed to researching more sensitive methods.
5. In Point-of-Care Testing section, all methods seem optical. Are there some examples employing electrochemical detections for POCT applications?
Response & Revisions: Thanks for the reviewer’s comments and suggestions. We added the POCT-based electrochemical methods in 4.1.POCT based detection of enzyme activity/proteins/compound (line 434-448, p.15) and 4.2.POCT based detection of CTC (line 468-481, p.16).
6. Texts in Figure 3 are too small. I can’t see texts of panel D even if 200% magnified.
Response & Revisions: Thanks for the reviewer’s comments. We have modified Figure 3 and increased the font size.

Reviewer 2 Report
The authors report a review regarding the detection of circulating tumor cells (CTC) using electrochemical methods. The ms reviews at the beginning the main physical methods for detection of CTC then it focused on the electrochemical detection. Overall, this is a very timely, interesting and comprehensive review which provides a nice overview of the importance of CTC and its detection. The ms is also well written and well organised. My only recommendation is to suggest the authors to add a table summarizing the performances on CTC detection using the physical methods. Apart from that, this is a very nice review.
Author Response
Our response could also be found in the attachment.
Comments:
The authors report a review regarding the detection of circulating tumor cells (CTC) using electrochemical methods. The ms reviews at the beginning the main physical methods for detection of CTC then it focused on the electrochemical detection. Overall, this is a very timely, interesting and comprehensive review which provides a nice overview of the importance of CTC and its detection. The ms is also well written and well organised. My only recommendation is to suggest the authors to add a table summarizing the performances on CTC detection using the physical methods. Apart from that, this is a very nice review.
Response & Revisions: We thank the reviewer for the positive assessment. We added a new Table 1 (on p.5), summarized the performances of biological and physical properties-based CTC detection methods in detail. Thanks.

Reviewer 3 Report
The manuscript reviews recent achievements in detection cancer cells by electrochemical biosensors based on nucleic acid aptamers. It is in scope of the Sensors. However, novelty is not convincing considering that recently more detailed review on similar topic has been published in Biosens. Bioeelectr. 2020. Manuscript should be substantially improved prior publication. In particularly:
- Provide sensitivity of the separation of CTC (in cells/mL) based on biological and physical properties.
- Ln 110 the Young modulus of CTC is indicated in the range 560-2472 Pa. Why so wide range and how the values differ form various CTC.
- In the Introduction include information about the structure and composition of the CTC, provide picture such as scheme, AFM or SEM image. Mention what typical caner markers can be found depending on the type of cancer.
- Provide the scheme summarizing various electrochemical methods of detection CMC using aptamers as receptors.
- Provide table summarizing the sequences of aptamers used for detection CTC. Include basic properties of aptamers, such as Kd and for which cancer markers the respective aptamer was selected.
- label-free electrochemical detection of leukemia cells was reported also in paper published in Electroanalysis 2018, 30, 1487-1495. This should be mentioned.
- Discuss the results in Table 1, such as advantages and disadvantages of various approaches.
- Very detailed review on the detection of CTC has been recently published by Safarpour et al. Biosens. Bioelectr. 2020, 148, 111833. It is therefore not clear what is novelty of the manuscript. I would recommend extending review also on acoustic methods for detection tumor cells. See for example Lecture Notes in Electrical Engineering, 2018, 464, 46-55.
9. Show references in the text in brackets, such as [1,2], etc, not as a subscripts
Author Response
Our response could also be found in the attachment.
Comments:
The manuscript reviews recent achievements in detection cancer cells by electrochemical biosensors based on nucleic acid aptamers. It is in scope of the Sensors. However, novelty is not convincing considering that recently more detailed review on similar topic has been published in Biosens. Bioeelectr. 2020. Manuscript should be substantially improved prior publication. In particularly:
Our response: We thank the referee for these comments. The review published in Biosensors and Bioelectronics is indeed a more detailed and comprehensive review of CTC-based electrochemical detection. But the novelty of our work is as follows: (1) The electrochemical methods summarized in our review are mainly the latest methods published in 2020. Besides, we paid more attention to clinical application, and proposed the existing problems and possible solutions. (2) We introduced the Point-of-Care testing (POCT), as a sensitive, fast, cheap, easy-to-operate method, it allows patients to realize sample input and result output, and it doesn’t need for complex equipment. It could greatly popularize the clinical application of CTC testing. And we discussed some possible combinations of POCT and CTC detection. As far as we know, our review discussed the possibility of combining POCT and CTC for the first time.
1. Provide sensitivity of the separation of CTC (in cells/mL) based on biological and physical properties.
Response & Revisions: Thanks for the reviewer’s suggestion. We added a new Table 1 (on p.5), summarized the performances of biological and physical properties-based CTC detection methods. For these methods, sensitivity is generally expressed in terms of capture efficiency (%) instead of cells/mL. The specific meaning of capture efficiency is the number of cells that can be detected divided by the number of cells added. Most methods can reach the sensitivity of 1 cell/mL, and some methods may have higher sensitivity, 0.5 cell/mL [1].
2. Ln 110 the Young modulus of CTC is indicated in the range 560-2472 Pa. Why so wide range and how the values differ form various CTC.
Response: Due to different measure methods including atomic force microscope (AFM), micropipette aspiration, microfluidic device, etc., different cell handling and cell heterogeneity, the measured results of Young's modulusmay be various. To make the results consistent, we listed the Young's modulus of cancer cells measured by AFM, which are 494-2472 Pa (There is an error in the original manuscript, we have corrected it). More precisely, the Young's modulus of ovarian cancer cell is 494 ± 222 Pa (HEYA8), 884 ± 529 Pa (HEY) and 2472 ± 2048 Pa (IOSE) [2], and the Young's modulus of lung cancer cell, breast cancer cell and pancreatic cancer cell is 520 ± 120 Pa, 500 ± 80 Pa and 540 ± 80 Pa [3], respectively. We can see that due to the different types of cancer, the heterogeneity between different cell lines even under the same cancer type, the Young's modulus varies greatly.
3. In the Introduction include information about the structure and composition of the CTC, provide picture such as scheme, AFM or SEM image. Mention what typical caner markers can be found depending on the type of cancer.
Response & Revisions: Thanks for the reviewer’s suggestions. We added a paragraph in 2.1.Methods based on biological properties (line 96-117, p.3-4). Including the process of metastasis, the origin of CTC, some basic biological properties of CTC, and typical cancer markers used for different types of cancer. In addition, we added a new Figure 2 (p.4), showed the typical fluorescence image and SEM image of CTC.
4. Provide the scheme summarizing various electrochemical methods of detection CMC using aptamers as receptors.
Response: We have summarized various electrochemical methods in 3.Electrochemical detection of CTC. Detection method based on current change such as differential pulse voltammetry (DPV), square wave voltammetry (SWV), linear sweep voltammetry (LSV) and Cyclic voltammetry (CV). Detection method based on resistance change such as EC impedance spectroscopy (EIS). Besides, electrochemiluminescence (ECL) and photoelectrochemical (PEC) are also included.
5. Provide table summarizing the sequences of aptamers used for detection CTC. Include basic properties of aptamers, such as Kd and for which cancer markers the respective aptamer was selected.
Response & Revisions: Thanks for the reviewer’s suggestions. We added a new Table 2 (p.7), summarized common aptamers, and listed their Kd values, targeted biomarkers, applicable cell lines and cancer types, etc.
6. label-free electrochemical detection of leukemia cells was reported also in paper published in Electroanalysis 2018, 30, 1487-1495. This should be mentioned.
Response & Revisions: Thanks for the reviewer’s comments. We added the label-free electrochemical detection of leukemia cells in our paper (line 246-249, p.8).
7. Discuss the results in Table 1, such as advantages and disadvantages of various approaches.
Response & Revisions: Thanks for the reviewer’s comments. We added a discussion about original Table 1 (current Table 3), including the advantages and disadvantages of various approaches (line 310-313, p.11).
8. (1) Very detailed review on the detection of CTC has been recently published by Safarpour et al. Biosens. Bioelectr. 2020, 148, 111833. It is therefore not clear what is novelty of the manuscript. (2) I would recommend extending review also on acoustic methods for detection tumor cells. See for example Lecture Notes in Electrical Engineering, 2018, 464, 46-55.
Response & Revisions: Thanks for the reviewer’s comments and suggestions. (1) For the novelty of this review, we have responded at the beginning (the response to comments). In conclusion, we introduced the Point-of-Care testing (POCT), summarized the current POCT-based CTC detection and discussed some possible development directions. As far as we know, our review discussed the possibility of combining POCT and CTC for the first time. In addition, for electrochemical-based CTC detection, we summarized the latest published literature. We paid more attention to the clinic use, and proposed the existing problems and possible solutions.
(2) We also extended our review on acoustic-based CTC detection according to the suggestions (line 168-180, p.6).
9. Show references in the text in brackets, such as [1,2], etc, not as a subscripts
Response & Revisions: Thanks for the reviewer’s suggestion. We have revised our article and showed all references in brackets.
References:
[1] Cote, R. J.; Datar, R. H., Size-Based and Non-Affinity Based Microfluidic Devices for Circulating Tumor Cell Enrichment and Characterization. 2016, 10.1007/978-1-4939-3363-1 (Chapter 3), 29-45
[2] Xu, W. W.; Mezencev, R.; Kim, B.; Wang, L. J.; McDonald, J.; Sulchek, T., Cell Stiffness Is a Biomarker of the Metastatic Potential of Ovarian Cancer Cells. Plos One 2012, 7 (10).doi:ARTN e4660910.1371/journal.pone.0046609
[3] Cross, S. E.; Jin, Y. S.; Rao, J.; Gimzewski, J. K., Nanomechanical analysis of cells from cancer patients. Nat Nanotechnol 2007, 2 (12), 7803.doi:10.1038/nnano.2007.388

Round 2
Reviewer 3 Report
The authors properly addressed the reviewers comments. Paper is now suitable for publication in Sensors.